# Quality of Life in the Postpartum Recovery of Women Who Practice Hypopressive Exercise: Randomized Clinical Trial

**DOI:** 10.3390/jcm11195592

**Published:** 2022-09-23

**Authors:** Juan Carlos Sanchez-Garcia, María José Aguilar-Cordero, Maria Montiel-Troya, Ana Eugenia Marín-Jiménez, Jonathan Cortes-Martin, Raquel Rodriguez-Blanque

**Affiliations:** 1Andalusian Plan for Research, Development and Innovation, CTS 1068, 18014 Granada, Spain; 2Andalusian Plan for Research, Development and Innovation, CTS 367, 18014 Granada, Spain; 3Department of Nursing, Faculty of Health Sciences, University of Granada, 18016 Granada, Spain; 4Department of Nursing Ceuta Campus, Faculty of Health Sciences, University of Granada, 51001 Ceuta, Spain; 5Quantitative Methods for the Economics and Enterprise, Faculty of Economics and Business Sciences, University of Granada, 18071 Granada, Spain; 6San Cecilio Clinical Hospital, 18071 Granada, Spain

**Keywords:** physical activity, exercise, hypopressive exercises, postpartum period, quality of life, HRQoL

## Abstract

Background: Practicing physical exercise is a way of maintaining a healthy lifestyle and it has been demonstrated to provide benefits and improve quality of life after labour. Aim: The objective of this study is to determine if performing a moderate hypopressive exercise program after childbirth improves the HRQoL of women. Materials and Methods: A 12-week randomized clinical trial was carried out. The women in the exercise group followed a training program based on the hypopressive method, which comprises postural and respiratory exercises of moderate intensity. We used the SF-36v2 questionnaire to assess the Health-Related Quality of Life in women. Results: We found statistically significant differences (*p* < 0.05) in general health components, vitality, emotional role, and mental health in the sample group of exercise group of women, who obtained a higher score. We found statistically significant differences in all aspects between the groups. The interaction between time and group affect in general health components, vitality, emotional role and the score of mental components. In all of them, the score was higher at the end of the programme and in the exercise group. Conclusions: The training program with hypopressive physical exercise improves the quality of life after childbirth. According to the guidelines recommended by the ACOG, the hypopressive method is valid for the improvement of HRQoL in women who perform this method at 16 weeks of delivery. Further studies are recommended to start such exercises one week after delivery in postpartum women.

## 1. Introduction

Health-Related Quality of Life (HRQoL) is defined by Urzúa [1] as “the well-being level obtained from the self-assessment of diverse domains of someone’s life, considering the impact that these domains have on his/her own health state”. The use of this concept is especially important when describing the impact of a disease in the life of patients and in the assessment of the efficacy of treatments.

The American College of Sports Medicine (ACSM) [2], in its last edition about physical exercise prescription, recommends practicing 150 min of weekly exercise in adults, spreading the physical activity among most days of the week, in order to improve the cardiovascular and central nervous systems. Currently, the American College of Obstetricians and Gynecologists (ACOG) [3] recommends regular exercise during pregnancy and after childbirth, in the absence of medical and obstetric complications, at least three times a week for 30–40 min, preferably a continuous activity rather than an intermittent one [2]. After labor, physical exercise not only helps women recover their pregestational weight, but also improves mental health, generates positive feelings, and reduces depression [4,5]. According to the ACOG in its document AP131 notes that exercise helps strengthen and tone abdominal muscles, boosts energy, helps prevent postpartum depression, promotes better sleep, relieves stress, and can help you lose extra weight gained during pregnancy [6].

ACOG [3] recommendations have shown that abdominal strengthening exercises, including abdominal contraction exercises and breathing exercises, in which a maneuver increases the abdominal pressure pulling of the muscles of the abdominal wall, decreases the incidence of diastasis of the rectus abdominis and decreases the space between interrects in women that had a vaginal delivery or caesarean section, aimed at improving muscle tone of the pelvic area and abdomen.

For all these reasons, we hypothesize that, by performing a program of hypopressive exercises of a moderate nature, developed from week 16 to week 28 after delivery, will be associated with a better score in the SF-36v2 Health Questionnaire, following the guidelines of the ACOG recommendations for exercise during the puerperium.

The SF-36v2 is a questionnaire of 36 questions where positive and negative aspects of health are valued, that is used globally to assess the relationship between quality of life and health [7].

The objective of this study is to determine whether performing a moderate hypopressive exercise program after childbirth improves the HRQoL of women. 

## 2. Materials and Methods

### 2.1. Design

We designed an open-label randomized clinical trial in which both subjects and researchers knew about the treatments. We followed the CONSORT rules [8]. The trial is registered in the USA National Institute of Health (ClinicalTrials.gov) and titled “Physical Activity in Pregnancy and Postpartum Period, Effects on Women”. Number NCT02761967. This project was approved by the Ethical Committee for Research in Granada, under license number 2601.20.15.

All participants signed an informed consent form before the study, following the rules established by the Helsinki Declaration and reviewed by the World Medical Association regarding Informed Consent.

### 2.2. Subjects

The subjects were women whose babies were 14 weeks old, in March 2019. Recruitment was carried out at Healthcare Centers of the Granada-Metropolitan Healthcare District (SAS). We selected women that fulfilled the inclusion criteria of labor date, and they were later contacted by phone. The recruitment responsible researcher phoned the eligible subjects and provided them with verbal information. Those subjects who showed interest in participating in the project were sent an email that included an attached file with further information about the study.

### 2.3. Inclusion Criteria

The inclusion criteria involved having had a healthy pregnancy and an eutocic delivery; that is to say, the deliveries that have occurred without instruments for the expulsion of the infant. 

Another requirement was that prior to pregnancy they had not performed systematic physical exercise of a moderate nature.

### 2.4. Exclusion Criteria

Not signing the informed consent form was a condition for exclusion. Additionally, once the study ended, we excluded all subjects who attended less than 80% of the planned sessions.

### 2.5. Study

Between the weeks 14 to 16 of postpartum, we met the subjects to obtain a written informed consent. In the following individual meeting, during the same two weeks, we collected their anthropometric data and HRQoL questionnaire.

The study covered a 12-week period. We carried out three weekly sessions of 60 min each on Mondays, Wednesdays, and Fridays. Each session consisted of three phases. The first one was a warm-up session of the muscles involved in the workout. In the second or main phase, the subjects followed each of the postures of the hypopressive method, doing postural exercises with breathing exercises in a systematic way. The subject would experience decreased intra-abdominal pressure among other benefits, and these postural and breathing exercises are described by Rial T [9] (annex 1). The final phase consisted of stretching exercises and relaxation.

After 12 weeks, we collected the post-study data which again consisted of anthropometric data and a HRQoL questionnaire.

The subjects in the control group attended the regular appointments of postpartum control, just as the subjects in the exercise group did.

### 2.6. Variables and Assessment Tools

#### 2.6.1. Social-Demographic and Anthropometric Variables

The following variables were collected in the personalized meeting with the researcher in charge: subject age, marital status, height, weight at 16 weeks after labor (Weight1), weight at 28 weeks after labor (Weight2), parity, gestation time, education level, social class, previous physical activity and breastfeeding.

Weight (Kg) was assessed with a calibrated scale and height (m) was measured with a calibrated metal stadiometer. To calculate body-mass index (BMI) we used the formula BMI = weight (Kg)/height^2^ (m^2^) [10,11,12]. We classified the subjects following the World Health Organization (WHO) classification regarding nutritional states, in accordance with BMI, in the following categories: underweight <18.50 Kg/m^2^, normal weight 18.50–24.99 Kg/m^2^, overweight 25–29.99 Kg/m^2^, and obesity ≥30.00 Kg/m^2^.

Subjects’ social class was classified using a questionnaire of social class assessment in health sciences by Álvarez–Dardet et al., 1995 [13], which ranks subjects in five social groups according to their working abilities. 

Previous physical activity was assessed at postpartum week 16 according to the subjects’ personal report to the question “a typical day of a typical week” using the tool Global Physical Activity Questionnaire (GPAQ, from WHO “Global recommendations on physical activity for health”) [14].

#### 2.6.2. Level of Effort and Workout Intensity 

We used the classical Borg Scale of Perceived Exertion, or Rating of Perceived Exertion (RPE) [15] in which scores between 12 and 14 are a “somewhat hard” mean moderate level.

Heart rate was measured at the end of each workout using the pulse oxymeter Quirumed OXYM2000 in all women that had an RPE higher than 14.

#### 2.6.3. Health Related Quality of Life (HRQoL)

We used the self-administered SF-36v2 questionnaire at weeks 16 and 28 after labor to evaluate HRQoL. It is a 4-week reminder, with 8 health state domains or aspects. These domains are Physical Function (PF), Physical Role (PR), Body Pain (BP), General Health (GH), Vitality (VT), Social Function (SF), Emotional Role (ER), and Mental Health (MH). We included Physical Components Summary (PCS) and Mental Components Summary (MCS) [16] measurements.

### 2.7. Sample Size

This study is a component of a main project registered at Clinical Trials, hence the sample size was calculated for the main project according to previous studies by Barakat et al. (2011) [17]. In their study, Barakat et al. followed a program of physical exercises with pregnant women from gestation weeks (GW) 6–9 until GW 38–39. They found that the percentage of women with a perception of good/very good health reached 96.9% in the treatment group, compared to 81.8% in the control group. In order to achieve 80% power when detecting differences in a null hypothesis test, H0: p1 = p2 using a bilateralChi-square test for two independent samples and considering a significance level of 5%, our sample should be of 56 subjects per group, or 112 in total. In this study, the sample size for the control group was 65 women, and the sample size for the exercise group was 64 women, a total of 129 subjects. 

### 2.8. Randomization

The sample allocation was randomized, following an open-label technique of simple and randomized sampling without replacement, where both subjects and researchers knew about the treatments. Once the participants were selected at the Healthcare Center and after verifying that they met the inclusion criteria, we introduced copies of the numbers assigned to the subjects in a vase. The main researcher of the randomized clinical trial (RCT) extracted the numbers and assigned firstly the control group members (“*n*” = 65) and then the exercise group (*n* = 64). The women randomly assigned to the exercise group had an interview with the main researcher, who informed them of the benefits of attending the sessions and provided them with the documents explaining the treatment.

### 2.9. Statistical Analysis

We ran a descriptive analysis of the main studied variables per group.

We compared both groups using Student’s t-distribution for continuous variables and the Chi-Square Test for qualitative variables.

A factorial design with two factors with fixed effect was used to test the effect of group and time factor (data collect postpartum weeks 16–28). We used partial Eta-Squared to calculate the effect of factors. We considered Eta-Squared small effect 0.01, medium effect 0.06 and large effect 0.14.

We obtained the commercial license #QM035814 from the company Quality Metric Incorporated to analyze the SF-36v2 Health Questionnaire, and we studied the questionnaire variables in independent groups. 

We used SPSS statistical package for Social Sciences Software (version 19 for Windows, SPSS Inc., Chicago, IL, USA) to run the statistical analysis with a statistically significant level defined at <0.05.

## 3. Results

During the study we lost subjects in each group, mostly due to subjects returning to their working schedules, the final sample size being 119 subjects. The final sample size for the exercise group was 56 women, and for the control group 63 women. Figure 1 shows the flowchart of the sample. Table 1 shows the basal characteristics of the sample.

No statistically significant differences were found in baseline characteristics between the exercise (*n* = 65) and control groups (*n* = 64). 

The women had a mean age of 32.12 ± 4.43 in the exercise group and 30.58 ± 4.75 in the control group. About 45% of women in both groups were married. BMI ranged from 23.89 for those in the exercise group to 24.01 for the control group. 

Regarding the physical activity prior to the study, collected with the GPAQ instrument, the daily activity carried out by the women in general was moderate, as was movement and recreational activity.

The practice of breastfeeding in postpartum women was studied and no statistically significant results were observed.

With the purpose of knowing if group, data collection week and the interaction between week and group affect each of the measured parameters of the survey, a two-factor fixed effect factorial design was used.

Table 2 shows the average and typical deviation for each aspect per group, week and interaction between week and group, *p*-value for each comparison and factor effect in the value of each aspect. 

As shown in Table 2, aspects of vitality (VT), emotional role (RE), mental health (MH) and mental component punctuation (MCS) are affected by the origin group, and in all cases the exercise group having a higher punctuation. In vitality (VT), emotional role (RE) and mental health (MH) the effect of group over punctuation is small; for the sum of mental component punctuation (MCS) the variable group has a medium effect. Time affects all variables, except body pain (BP). In all cases, punctuation is higher at week 28. At week 16 postpartum, there are several aspects with average score under 50, being emotional role (RE) the lowest, with a mean of 28.71 points. Also in the lower punctuation range are emotional role (RE) and vitality (VT), possibly implying that women experience certain physical limitations 4 months after delivery. After 12 weeks, at week 28 postpartum, in basically all aspects, score mean punctuations are above 50, except emotional role (RE), which increases to a mean punctuation of 41.46. The effect of time on physical component (PF) is small. For the social function (SF) and physical component punctuation, it could be considered medium. For physical role (RP), general health (GH) vitality (VT), emotional role (RE), mental health (MH) and sum of mental components punctuation (MCS), the effect is higher, with emotional role (RE) being the highest of all.

The effect of group–week interaction is significant for general health (GH), vitality (VT) emotional role (RE) and mental component punctuations (MCS). Table 2 shows the mean for each according to group and week, being in all cases higher at week 28, particularly for Exercise group. The effect of interaction is small for general health (GH) and emotional role (RE), while medium for vitality (VT) and sum of mental component punctuation (MCS). Figure 2 shows the estimated population marginal means for each group and week, with a distinct line for each group, standing out significantly in GH, VT, RE and MCS at the end of the intervention women in the exercise group who improved the scores much more than those who did not perform physical exercise in the same period.

The effect of time and group and their interaction with each category according to body mass index (BMI) has been studied for physical components punctuation (PCS) and mental component punctuation (MCS). The results of this analysis, with means and standard deviations for each group, can be found in Table 3.

Group affects physical component punctuation (PCS) in women with obesity, showing a higher punctuation in the exercise group. Punctuation for mental components (MCS) is affected by group in the category of overweight women, having an average punctuation higher than exercise group. In both cases the effect of group can be considered high.

Time factor affects all categories in both of the studied components except Obesity in physical component punctuation (PCS). In all cases punctuation is higher at week 28 postpartum. The effect of time can be considered medium for physical component punctuation (PCS) and large for mental component punctuation (MCS).

Interaction between time and group affects the mental component punctuation (MCS) in the case of normal weight women, with higher punctuation at week 28. The effect of this interaction is medium for Exercise group.

The effect of this interaction on each Body Mass Index (BMI) category is showed in Figure 3.

## 4. Discussion

In the study it has been observed a high follow-up rate that may be due to the mothers’ concern for the recovery of their pre-pregnancy fitness, issue to be studied. Our results Could be extrapolated to help other women with healthy pregnancies and eutocic deliveries.

Although using a self-administered questionnaire could be considered a limitation, Ware et al. (1993) [18] claimed that there were no differences in the internal consistency between the SF-36v2 questionnaire and interviewer-administered questionnaires. Attracting pregnant women to participate in the study was also a limitation, which we attribute to the lack of existing information in the Healthcare Services about the importance of physical activity for pregnant women. The need to reduce environmental barriers was an additional limitation we had to resolve, which we did by offering a daycare service to the women participating in the study.

Other limitations of the study were that it did not ask about the consumption of drugs and/or medications (caffeine, tobacco, alcohol, etc.), nor the presence of medical and psychiatric disorders (depression and postpartum anxiety), which could influence the variables studied.

Singh et al., 2015 [19] tried to determine the impact of parity in women’s HRQoL. They studied 60 women that met the inclusion criteria and found that women with a higher level of parity tend to have worse HRQoL. In our study we did not find statistically significant differences between groups regarding parity. However, we did find that HRQoL is inversely associated with sedentariness during the studied period, meaning that HRQoL is reduced as sedentariness increases.

Haas et al., 2005 [20], already confirmed that the lack of physical exercise, before, during, and after pregnancy, is associated with a worse state in all health indicators. Doya et al., 2013 [21], examined the effect of specific workouts of physical training during pregnancy, particularly ones that focus on the core, on QoL at the end of pregnancy and during the 2 months after labor. They studied a sample size of 49 nulliparous women between pregnancy weeks 24 and 36 and the treatment group carried out a structured training program centered on trunk fitness. They measured HRQoL using questionnaire SF-36v2 at pregnancy week 24 and 2 months after labor. They found that all the SF-36v2 domains were more favorable in the training group two months after labor than in the control group. Their results are similar to the ones we obtained in our study using the SF-36v2 questionnaire at six months after labor.

Haruna et al. (2013) [22] carried out a clinical trial studying a workout program at three months after labor, consisting on a 4-week program of 90 min of exercise per week. They found that the workout program for healthy women during the postpartum period improved their HRQoL and self-esteem. However, in that study they only found a statistically significant difference between the domains PF and VT, but not between PCS and MCS. Their results are in contrast with the results of our study, in which from the fourth to the sixth month postpartum we found statistically significant differences in the SF-36v2 domains GH, VT, SF, ER and MH. We reach the same conclusion as Haruna et al., confirming that physical exercise contributes to improvement of HRQoL.

In a systematic review of the scientific literature published by our research group [23], we concluded that supervised workouts of moderate intensity and with a longer duration than six weeks yields statistically significant results in HRQoL, which we also confirm in our clinical study. Additionally, in our study we found the exception that categorizing women on both groups by BMI creates a statistically significant increment of the recorded scores in the SF-36v2 questionnaire of the PCS and MCS in both groups.

Similarly, Yan and Chen [24], carried out a simple randomized blind pilot study with the objective of exploring whether aerobic workouts improve postpartum stress, sleep quality, fatigue, and depression. To measure their data, they used the perceived stress scale, postpartum fatigue scale, postpartum sleep quality scale, and the Edinburgh postnatal depression scale. They found results similar to ours, which can motivate women that gave birth to practice postpartum physical exercise to improve the studied parameters. In this same line of research, an additional study [25] was designed to study the improvement of the sleep quality associated to postpartum physical exercise.

## 5. Conclusions

In this study we found that offering a physical exercise program following the ACOG recommendations of the hypopressive exercise program to postpartum women significantly improves the HRQoL of these women. Control women improved their SF-36v2 questionnaire score. However, that improvement is normal just as the postpartum period advances, as they present scores that are clearly lower than women that exercise during the postpartum period.

Based on the above, the described study concludes that it is beneficial to recommend physical exercise following the hypopressive methodology in postpartum to any woman who has no contraindication to be able to perform physical exercise.

ANNEX 1 [9]: 



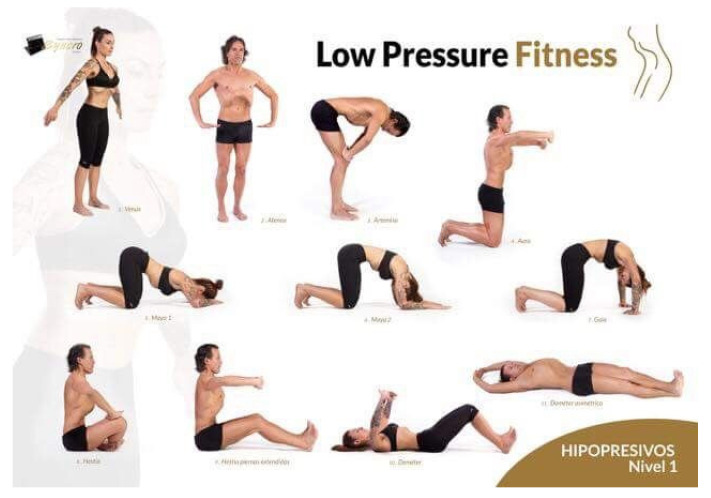



## Figures and Tables

**Figure 1 jcm-11-05592-f001:**
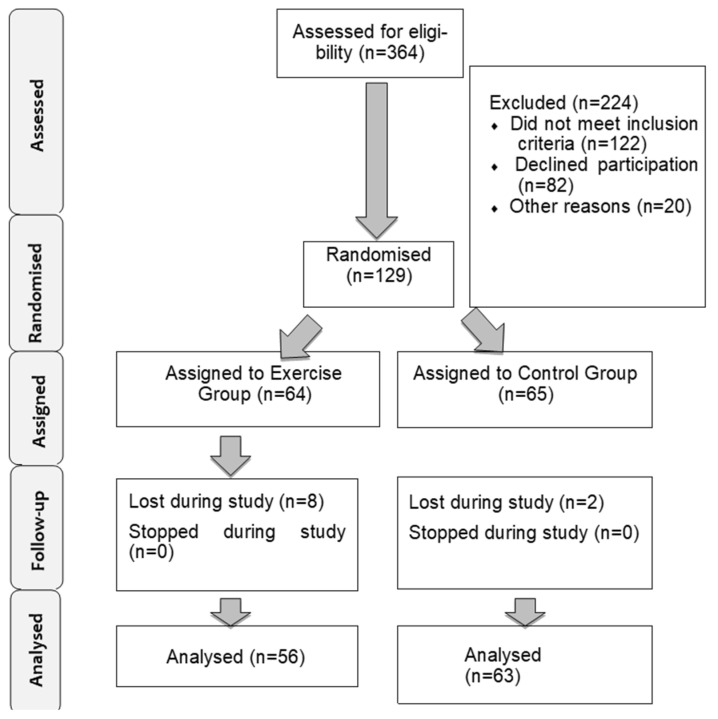
Description of the sample selection.

**Figure 2 jcm-11-05592-f002:**
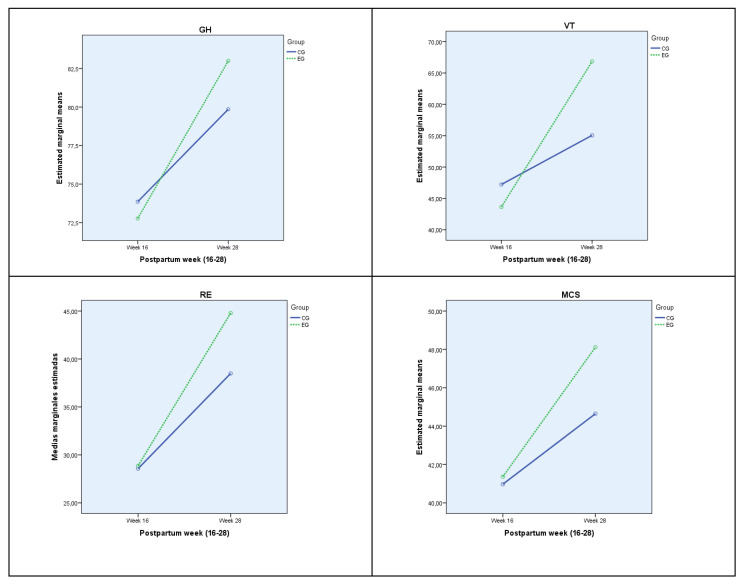
Population marginal means, estimated for each week for each group. GH: General Health; VT: Vitality; ER: Emotional Role; MCS: Mental Component Summary. EG: Exercise Group; CG: Control Group.

**Figure 3 jcm-11-05592-f003:**

Marginal population means estimated for each week for each MCS group according to BMI. MCS: sum of mental components punctuation; EG: Exercise Group; CG: Control Group.

**Table 1 jcm-11-05592-t001:** Basal characteristics of the sample.

Characteristics	Exercise Group (*n* = 65)	Control Group (*n* = 64)	*p*-Value
Age in years (Mean ± SD)	32.12 ± 4.43	30.58 ± 4.75	0.331
min–max	21–43	22–43	
Marital Status (Freq. (%))	57 (44.2)	59 (45.7)	0.396
Height (Mean ± SD)	1.646 ± 0.06	1.651 ± 0.05	0.604
Weight1 (Mean ± SD)	67.07 ± 12.23	67.89 ± 12.58	0.71
BMI 1 (Median [Q1–Q3])	23.89 [21.52–27.51]	24.01 [21.78–26.58]	0.953
Normal weight(Mean ± SD)	19 (29.2)	12 (18.8)	0.322
Overweight (Mean ± SD)	31 (48.4)	30 (46.2)	
Obesity (Mean ± SD)	21 (32.8)	16 (24.6)	
Parity (Freq. (%))	20 (30.77)	17 (26.56)	0.739
Gestation Time in days(Mean ± SD)	280.09 ± 8.257	279.70 ± 8.919	0.996
**Physical Activity Domains (according to intensity) GPAQ**			
Daily Activity			0.880
Moderate	57 (93.4)	56 (94.9)	
Vigorous	4 (6.6)	3 (5.1)	
Moderate Movement	0.497	44 (68.8)	0.497
Recreational Activity			
Moderate	52 (80.0)	53 (82.8)	0.681
Vigorous	13 (20.0)	11 (17.2)	

BMI: Body Mass Index; GPAQ: Global Physical Activity Questionnaire.

**Table 2 jcm-11-05592-t002:** Average score (SD) per group, postpartum week and interaction group * week. (EG, *n* = 56; CG, *n* = 63).

Aspect	Group				Postpartum Week				Interaction Group * Week					
									LPF		Seden		*p*	η^2^
	EG	CG	p	η2	16 Post	28 Post	p	η2	16 Post	28 Post	16 Post	28 Post		
PF	75.54 (14.40)	74.56 (14.27)	0.597	0.001	72.52 (15.46)	77.52 (12.64)	0.007	0.031	72.86 (14.95)	78.21 (13.43)	72.22 (16.01)	76.90 (11.96)	0.855	0.000
RP	44.87 (16.40)	43.95 (13.44)	0.611	0.001	38.97 (14.84)	49.79 (12.86)	<0.001	0.137	38.06 (15.55)	51.67 (14.38)	39.78 (14.26)	48.12 (11.20)	0.144	0.009
BP	60.27 (17.53)	60.46 (12.71)	0.922	0.000	58.55 (14.89)	62.19 (15.22)	0.054	0.016	57.21 (16.75)	63.32 (17.90)	59.73 (13.04)	61.19 (12.44)	0.237	0.006
GH	77.88 (9.60)	76.86 (7.63)	0.298	0.005	73.34 (8.78)	81.34 (6.30)	<0.001	0.225	72.77 (9.67)	83.00 (6.26)	73.86 (7.95)	79.86 (6.01)	0.033	0.019
VT	55.25 (17.25)	51.14 (14.56)	0.020	0.023	45.54 (14.94)	60.61 (13.23)	<0.001	0.252	43.64 (14.08)	66.85 (11.30)	47.22 (15.59)	55.06 (12.39)	<0.001	0.076
SF	83.37 (11.79)	81.15 (12.92)	0.158	0.008	79.31 (13.26)	85.08 (10.83)	<0.001	0.057	79.46 (12.25)	87.28 (9.97)	79.17 (14.20)	83.14 (11.26)	0.221	0.006
RE	36.83 (12.64)	33.53 (9.69)	0.006	0.032	28.71 (9.19)	41.46 (9.42)	<0.001	0.340	28.87 (9.40)	44.79 (10.24)	28.57 (9.07)	38.49 (7.55)	0.012	0.027
MH	78.30 (10.45)	75.32 (9.89)	0.018	0.024	73.53 (10.94)	79.92 (8.41)	<0.001	0.101	74.73 (11.46)	81.88 (7.95)	72.46 (10.43)	78.17 (8.49)	0.570	0.001
PCS	48.28 (5.15)	48.47 (4.16)	0.749	0.000	47.37 (4.71)	49.39 (4.36)	0.001	0.050	46.89 (4.96)	49.67 (5.00)	47.80 (4.48)	49.15 (3.73)	0.228	0.006
MCS	44.74 (5.16)	42.81 (3.97)	<0.001	0.064	41.16 (4.25)	46.28 (3.51)	<0.001	0.333	41.36 (4.66)	48.11 (2.99)	40.97 (3.89)	44.65 (3.13)	0.002	0.042

PF: Physical Funtion; RP: Physical role; BP:Body pain; GH: General healthl; VT: Vitality; SF: Social function; RE: Emotional role; MH: Mental health; PCS: Physical component summary; MCS: Mental component summary; EG: Exercise Group; CG: Control Group. η^2^ = 0.10: small effect; η^2^ = 0.30: medium effect; η^2^ = 0.50: big effect.

**Table 3 jcm-11-05592-t003:** Physical component summary and: Mental component summary average score (SD) per group, postpartum week and interaction week * group according BMI. (Normal Weight: EG, *n* = 32, CG, *n* = 34; Overweight: EG, *n* = 17, CG, *n* = 22; Obesity: EG, *n* = 7, CG, *n* = 7).

Aspect	BMI	Group				Postpartum Week				Interaction Week * Group					
										EG		CG		*p*	η^2^
		EG	CG	*p*	η^2^	16 post	28 post	*p*	η^2^	16 post	28 post	16 post	28 post		
PCS	Normal Weight	48.06 (4.40)	49.31 (4.50)	0.103	0.021	47.83 (4.53)	49.58 (4.28)	0.022	0.041	46.79 (4.02)	49.33 (4.46)	48.81 (4.82)	49.81 (4.16)	0.315	0.008
	Overweight	48.01 (6.41)	48.35 (3.26)	0.757	0.001	46.85 (4.96)	49.55 (4.43)	0.011	0.084	46.11 (6.31)	49.90 (6.12)	47.42 (3.65)	49.27 (2.58)	0.373	0.011
	Obesity	49.96 (4.97)	44.78 (2.91)	0.003	0.310	46.66 (4.98)	48.07 (4.66)	0.382	0.032	49.27 (5.27)	50.64 (4.96)	44.05 (3.17)	45.50 (2.67)	0.979	0.000
MCS	Normal Weight	43.82 (4.94)	43.37 (4.33)	0.491	0.004	40.96 (4.30)	46.21 (3.25)	<0.001	0.343	40.29 (4.15)	47.35 (2.56)	41.59 (4.40)	45.15 (3.49)	0.008	0.054
	Overweight	46.95 (4.79)	42.51 (3.12)	<0.001	0.320	42.36(4.01)	46.55(3.97)	<0.001	0.311	44.15 (4.54)	49.77 (3.15)	40.97 (2.96)	44.06 (2.48)	0.099	0.036
	Obesity	43.57 (5.75)	50.00 (4.13)	0.062	0.137	38.73 (3.73)	45.83 (3.55)	<0.001	0.548	39.96 (4.75)	47.63 (3.24)	37.96 (2.47)	44.03 (3.04)	0.445	0.025

PCS: Physical component summary; MCS: Mental component summary; EG: Exercise Group; CG: Control Group. η^2^ = 0.10: small effect; η^2^ = 0.30: medium effect; η^2^ = 0.50: big effect.

## Data Availability

Available upon request from the corresponding author.

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
