# Peer review of "Quality of Life in the Postpartum Recovery of Women Who Practice Hypopressive Exercise: Randomized Clinical Trial"

_jcm, 2022, doi:10.3390/jcm11195592_

Round 1

Reviewer 1 Report

Dear Authors

Please note the following

it should be clarified the diversity of exercises between groups of women, e.g. women after cesarean section or normal delivery, women with pregnancy pathology, hypertension, diabetes...

also, it is important for the reader to mention why exercise improves the quality of life.  Do endorphins contribute?

replace the word "baby" with the word "infant"

what information did we have about these women?

what role did exercise play in their lives before pregnancy?

the basic characteristics of the samples, as long as they are not statistically significant, can simply be mentioned

strengths and limitations should be placed at the end of the discussion.

the postpartum period is a transitional period in a woman's life, which, apart from physical and hormonal changes, also includes changes in the woman's mental state. The quality of life of a woman, therefore, depends on many factors which have not been sought in this article and unfortunately do not lead us to safe conclusions.

Author Response

Dear Reviewer, thank you very much for your report and your assessments. We have tried to adjust the article to your requests as best as possible. If you think we need to make more changes, let us know. Best regards, María.

Reviewer 2 Report

The review of manuscript entitled “Quality of life in the postpartum recovery of women who practice hypopressive exercise: randomized clinical trial”

The aim of this study was to investigate the influence of physical exercises on Heath-related quality of life after labour. It was  a 12-weeks randomized clinical trial,  the SF-36v2 questionnaire was used  to assess the Health-Related Quality of Life.

My comments:

1.      The trial was registered in the USA National Institute of Health (ClinicalTrials.gov), therefore there is no point in commenting on its assumptions, they are correct, unfortunately the publication itself seems to be partial and probably covers only a small part of the actual research

2.      the summary is too general, he took the numbers, after reading it the reader knows nothing about the clinical trial, more details

3.      American College of Obstetricians and Gynecologists (ACOG)  recommends regular exercise during pregnancy and after childbirth, so therefore we can expect that these exercises affect not only the muscles and statics of the reproductive system but, as in any other case, the quality of life, are the assumptions of the research not too obvious?

4.      Please explain why women were qualified for the study approximately 3 months after childbirth and not earlier?  The intensity of the exercise was very low, therefore there are no medical contraindications for women who have not undergone cesarean section

5.      Table 2 and 3 requires re-reading, it is completely unreadable, and the same for Figure 2 and 3

6.      “The high number of participants in this study, including the high follow-up rate, 242

gives strength to this study” I think the groups were rather small and not big? I recommend less categorically

7.      the discussion is very brief and cursory, citing little research and publications from several years ago. There are more recent studies available on these issues and it is properly structured, it is worth the search, the discussion needs to be improved

8.      ’’Based on the above, the described study concludes that it is beneficial to recommend physical exercise following the ACOG methodology in postpartum to any woman who has no contraindication to be able to perform physical exercise” -  so, the conclusions are that the authors of these recommendations are right, maybe it is worth formulating more innovative conclusions, somehow different, more politically?

9.      of course, these comments do not change the fact that the manuscript is interesting and deserving of publication, but requires significant changes

Author Response

(The authors gave the same response as above.)

Reviewer 3 Report

The authors designed an open-label randomized clinical trial to determine whether a workout programme using physical exercise has an impact on the Health-Related Quality of Life after labour.  First of all, I acknowledge the authors for presenting a topic that is so relevant and needs to be addressed by health services. Women's physical and mental health in the postpartum period has been understudied and underestimated by maternity services. Thus, the findings of the study contribute widely to the field of health sciences. Nevertheless, I think that some kind of revisions are needed before publication:  

a) Include reliability statistics for each questionnaire used. I suggest the calculation of the McDonald omega coefficient.

b) There is a typing error in Table 1 in the "Moderate movement" row for the "Exercise group" column. Also, clarify whether the GPAQ data are expressed as means or percentages.

c) Table 2 should be extensively improved. In this regard, there are words that are in Spanish; the title should be more explanatory; I suggest placing the eta-squared symbol instead of effect in column title; in the table footnote include the cut-off points with their interpretations of the effect size.

d) On line 214 the authors state "Table 2 shows the median for each according to group and week, being in all cases higher at week 28, particularly for Exercise group."...however the table displays averages/means not medians.

e) Figure 2 is of very low quality, making it very difficult to analyze. In addition, I suggest including in the figure caption the abbreviations used.

f) Figure 3 has words in Spanish language.

g) Although the results of marginal analysis are presented in Figures 2 and 3, there is a lack of development of these results in the text.

h) There are serious limitations that the authors should mention. The first limitation is that the study did not ask about the consumption of drugs and/or medications (caffeine, tobacco, alcohol, etc.) or the presence of medical and psychiatric disorders (especially postpartum depression and anxiety), which undoubtedly can influence all the variables evaluated. Nor are data presented on whether the participants practiced breastfeeding, and what type of breastfeeding they practiced (exclusive/partial). In this regard, it is known that breastfeeding is a factor that influences the anthropometry of postpartum women, as well as playing a protective role in mental health.

i) Grammar of the text should be revised and corrected.

j) What date was the study conducted?

k) Provide additional details on how missing data were addressed, if any.

Author Response

(The authors gave the same response as above.)

Round 2

Reviewer 1 Report

Dear Authors 

You managed to overcome all the problems!

Congratulations!

Reviewer 3 Report

All of my previous comments have been answered. I have no further suggestions. I recommend this manuscript for publication.